# Investigation of the therapeutic efficacy and resistance mechanisms of lytic phages targeting ST218 KL57 CR-hvKP

Liuqing Dou,[1] Jiayang Li,[2] Wenqi Wu,[2] Li Xu,[3] Mingjie Qiu,[3] Shuanghong Yang,[1] Jiajie Wang,[4] Sai Tian,[1] Zhitao Zhou,[3] Meilin Wu,[3] Yun Zhao,[5] Xiuwen Wu,[1,2] Jianan Ren[1,2]

**ABSTRACT** Carbapenem-resistant hypervirulent *Klebsiella pneumoniae* (CR-hvKP) infection is gradually increasing globally. Phage therapy is a viable application as an alternative to antibiotics. However, clinical application of phage therapy is restricted by phage resistance. To further explore the mechanism underlying phage resistance, particularly the difference observed between *in vivo* and *in vitro*, we employed a mouse intra-abdominal infection model to assess the antibacterial properties of two lytic phages and further isolate and characterize phage-resistant mutants. We identified that the majority of the mutation sites in the phage-resistant *K. pneumoniae* mutants were located in the capsular polysaccharide (CPS) gene cluster, as determined through genomic and transcriptomic analysis. However, some *K. pneumoniae* phage-resistant mutants, including RM01, RM02, and RM12, developed phage resistance by downregulating CPS and the respective transcriptional regulators without any mutations in the CPS gene. In summary, these findings provide further evidence supporting phage therapy, particularly addressing the issue of CR-hvKP infections.

**IMPORTANCE** The global rise in antibiotic resistance has rekindled interest in utilizing bacteriophage therapy as a potential solution. In this study, we explored the therapeutic potential of two novel bacteriophages, with a focus on their *in vivo* efficacy using mouse models, and analyzed the probable mechanisms of phage resistance in bacteria. Our results indicated that in a murine infection model, phages JLBP1001 and JLBP1002 for *Klebsiella pneumoniae* were highly effective, significantly improving mouse survival. We further characterized and analyzed phage-resistant *K. pneumoniae* isolated from the mice and found that the resistance mechanisms in an *in vivo* environment are primarily concentrated in the capsular polysaccharide gene cluster. In RM01, RM02, and RM12, *putA* contributes to phage resistance through point mutations. These insights are important for optimizing phage-based therapies, particularly in the context of multidrug-resistant bacterial infections.

**KEYWORDS** *Klebsiella pneumoniae*, phage therapy, phage resistance

*K*lebsiella pneumoniae, one of the "ESKAPE" pathogens, is a common pathogen for hospital-acquired infections globally (1). In recent years, due to the irresponsible utilization of antibiotics and the horizontal spread of resistance genes, multidrug-resistant bacteria have proliferated widely, with carbapenem-resistant hypervirulent *K. pneumoniae* (CR-hvKP) being one of the major clinical challenges currently faced (2–4). The virulence of *K. pneumoniae* primarily stems from the expression of certain virulence factors, including capsular polysaccharide (CPS), lipopolysaccharide, pili, siderophores, outer membrane proteins, and the type VI secretion system (5, 6), which enable the pathogen to resist recognition and destruction by the host immune system, thereby surviving and causing disease.

**Peer Reviewer** Ella R. Rotman, Northwestern University Feinberg School of Medicine, Chicago, Illinois, USA

Address correspondence to Jiayang Li, jiayanglinj@163.com, Xiuwen Wu, wuxiuwen@nju.edu.cn, or Jianan Ren, jiananr@nju.edu.cn.

Liuqing Dou and Jiayang Li contributed equally to this article. The author order was determined in order of increasing seniority.

The authors declare no conflict of interest.

See the funding table on p. 17.

As a type of virus, bacteriophages are microorganisms that mainly infect bacteria and fungi (7). As early as 1919, the microbiologist Félix d'Herelle utilized bacteriophages to treat bacterial dysentery (8). However, with the discovery and development of antibiotics, bacteriophage therapy gradually faded away. The option of using antibiotics has become more constrained in recent years due to the global increase in antibiotic resistance in bacteria. As increasing studies have reported the feasibility of phage therapy, phages are now being utilized to treat multidrug-resistant bacterial infections (9, 10).

In this study, two new types of phages were identified, and their efficacy was explored using a mouse intra-abdominal infection (IAI) model. Mutations in the CPS gene cluster were subsequently identified as one of the key mechanisms by which *K. pneumoniae* escapes phage infection under *in vivo* phage pressure. However, there are still some unknown mechanisms that resulted in the downregulation of the CPS gene expression level in several phage-resistant *K. pneumoniae* mutants, which resulted in phage resistance.

## RESULTS

### Isolation and characterization of two lytic phages

Two phages, named JLBP1001 and JLBP1002, were successfully recovered from hospital sewage. Both phages formed transparent plaques with a 2–3 mm diameter halo on a 0.75% Luria-Bertani (LB) agar plate after incubating overnight at 37°C (Fig. 1A). After prolonged incubation (24 h), the halo diameter expanded to 7–8 mm, suggesting that the two phages encode a polysaccharide depolymerase. Transmission electron microscopy (TEM) analysis disclosed that phage JLBP1001 has an icosahedral capsid (78 nm in diameter) with an elongated tail (approximately 195 nm × 10 nm) and attached tail fibers (Fig. 1B). In contrast, phage JLBP1002 exhibited a smaller icosahedral head (65 nm in diameter) and a shorter tail, with small tail fibers present. Both phages possessed tail fibers, a characteristic feature shared by viruses in the order *Caudovirales*.

An optimal multiplicity of infection (MOI) assay was conducted to determine the phage titer that resulted in maximal phage proliferation *in vitro*. The optimal MOI for phage JLBP1001 was 0.0001. When the MOI was less than 1, as the value decreased, the progeny phage gradually increased, reaching $4.8 \times 10^9$ plaque-forming units (PFUs)/mL. However, phage JLBP1002 displayed greater variation, producing a larger number of progeny phages at both MOIs of 10 and 0.001 (Fig. S2A). In terms of temperature stability, both phages maintained high stability under 4°C–50°C. When the temperature exceeded 60°C, the activity of both phages decreased to various extents (Fig. S2C). Additionally, both phages maintained good activity within a pH range of 4–10 (Fig. S2D). These results indicated that the two phages have excellent temperature and pH stability, which is beneficial for storage and transportation. Subsequently, the one-step growth curves of both phages under the condition of MOI = 0.1 are displayed in Fig. S2B. The curve showed that JLBP1001 had a 30-min latent period, a lysis period of 60 min, and an estimated burst size of about 17 PFUs/cell. In contrast, JLBP1002 had a 30-min latent period and a 40-min lytic period, and the burst size was approximately 124 PFUs/cell.

### JLBP1001 and JLBP1002 belong to two separate phage families

Gene analysis demonstrated that the two dsDNA phages were composed of 48,944 bp with a 50.49% GC content and 43,877 bp with a 54.15% GC content (Fig. S3A and B). The phage sequences were annotated using Pharokka (11), which identified 87 and 64 protein-coding sequences for JLBP1001 and JLBP1002, respectively. No genes associated with antibiotic resistance, virulence factors, or tRNA were detected in the genomes of either phage, and both phages were classified as virulent. These results indicated that the two phages are suitable candidates for phage therapy.

BLASTn (https://blast.ncbi.nlm.nih.gov/blast/Blast.cgi) was utilized to conduct separate database comparisons for JLBP1001 and JLBP1002, which obtained a total of 79

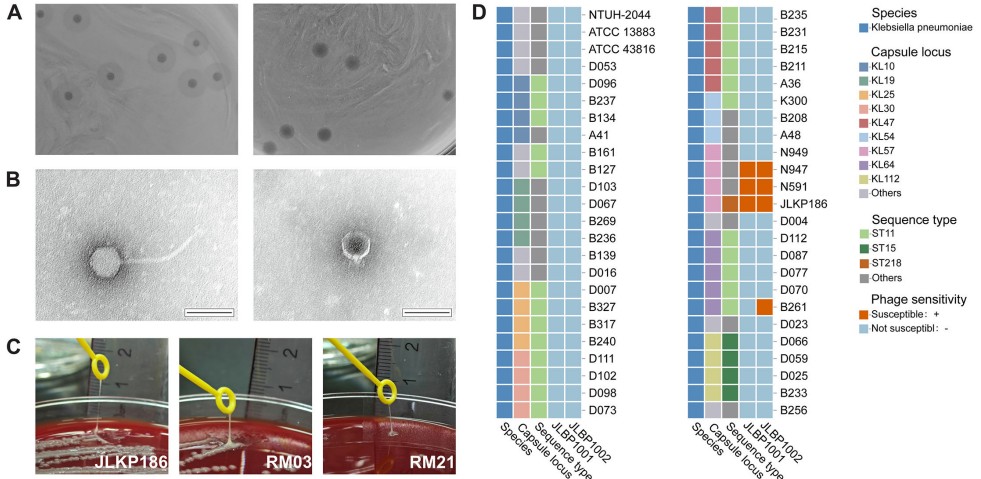

**FIG 1** The phenotypic characteristics of phages and host strains. (A) JLBP1001 (left) and JLBP1002 (right) with JLKP186 overnight culture formed clear-zone plaques. (B) TEM images of bacteriophages JLBP1001 (left) and JLBP1002 (right). Scale bar = 100 nm. (C) The positive string test of JLKP186, RM03, and RM21. (D) Schematic diagram of the host range of bacteriophages JLBP1001 and JLBP1002, with "others" in the "capsule locus" including KL01, KL02, KL09, KL15, KL20, KL21, KL53, KL63, KL107, and KL122 and "others" in the "sequence type" including ST17, ST23, ST29, ST111, ST258, ST320, ST489, ST493, ST592, ST617, ST1027, ST2237, ST2286, and ST5422; due to the relatively low proportion of the aforementioned KL serotype and ST type, they are classified under "others."

highly matched phages. Subsequently, VIRIDIC (12) phylogenetic analysis was conducted based on these publicly available, complete phage genomes. The results found that JLBP1001 had the highest similarity with *Klebsiella* phage RCIP0085 (94.3%) (Fig. S3C), while JLBP1002 had only 87.9% genomic similarity with *Klebsiella* phage RCIP0041 (Fig. S3D). Therefore, the two phage strains could be considered new species within their respective phage families. Viptree (13) analysis demonstrated that JLBP1001 and *Klebsiella* phage KP36 (NC_029099) were derived from the same branch, the *Drexlerviridae* family. JLBP1002 and *Klebsiella* phage KP2 (NC_028664) were members of the same *Autographiviridae* family branch (Fig. S3E and F).

## Two phages exhibit specificity for the KL57 serotypes of *K. pneumoniae*

Both phages shared the same host strain, JLKP186, an ST218 KL57 clinical isolate obtained from an inpatient with IAI. This strain displayed a hypermucoviscous phenotype, and the string test was positive (>5 mm) (Fig. 1C). Whole-genome sequencing (WGS) analysis found that JLKP186 carries the $bla_{KPC-2}$ gene, in addition to expressing five other virulence genes: *iucA*, *iroB*, *peg-344*, *rmpA*, and *rmpA2*. Mouse virulence tests confirmed JLKP186 as a hypervirulent strain. Additionally, antimicrobial susceptibility testing further revealed that JLKP186 was resistant to both imipenem and meropenem (minimum inhibitory concentration [MIC] ≥ 16 µg/mL), confirming it as a CR-hvKP (Table S1).

Subsequently, the host range of the two phages was assessed using 48 strains, which included a total of 19 KL serotypes, including KL57 (Table S2). Host range analysis demonstrated that JLBP1001 was only sensitive to KL57 serotypes, whereas JLBP1002 was sensitive to both KL57 serotypes and also lysed the ST11 KL64 strain, B261. Neither phage lysed the N949 (KL57) strain (Fig. 1D). Overall, the host range of phages JLBP1001 and JLBP1002 was relatively narrow, making them suitable for treating KL57 *K. pneumoniae* infections without disrupting the normal microbiota.

## Potent *in vitro* lytic activity of JLBP1001 and JLBP1002 against CR-hvKP

Figure 2A and B show the *in vitro* bacteriolytic activity of JLBP1001 and JLBP1002 at various MOIs. When the MOI ranged from 100 to 0.01, both phages completely

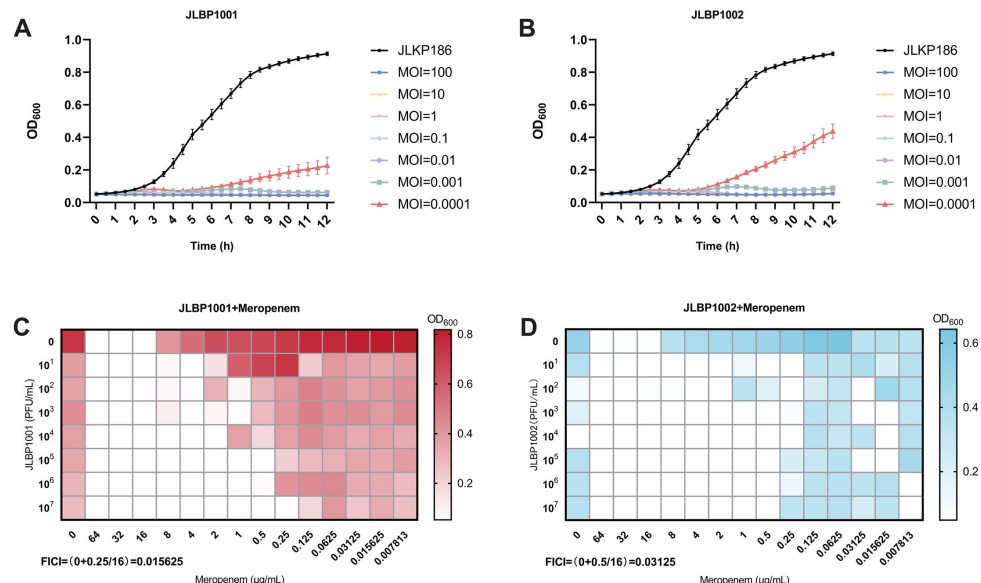

**FIG 2** The *in vitro* infection activity of JLBP1001 and JLBP1002. (A and B) Bacteriostatic effect of JLBP1001 and JLBP1002 against JLKP186 at different MOIs *in vitro*. Error bars indicate mean ± SD, and three replicates are performed for each experimental group. (C and D) Phage-antibiotic synergy (PAS) test of JLBP1001 and JLBP1002. The meropenem concentration at the abscissa ranges from 64 to 0.007813 µg/mL, the phage titer at the ordinate ranges from $10^1$ to $10^7$ PFUs/mL, and the growth inhibition is displayed using a heatmap. FICI was calculated by the formula: FICI = (MIC combination/MIC phage alone) + (MIC combination/MIC antibiotic alone). FICI ≤ 0.5 indicates a synergistic effect; 0.5 < FICI ≤ 1 indicates an additive effect; 1 < FICI ≤ 2 indicates no effect; and FICI > 2 indicates an antagonistic effect.

suppressed bacterial growth within 12 h. However, when the MOI was further reduced to 0.001 or below, bacterial regrowth was evident from 5 h onward and could no longer be inhibited by the phages. Although regrowth occurred later due to the emergence of resistant bacteria, the overall bacterial density remained lower than that of the positive control group.

To assess the potential efficacy of the combination of phages and antibiotics, we performed a PAS assay, which combined phages and meropenem (Fig. 2C and D). In the presence of meropenem, increasing the titer of JLBP1001 progressively decreased the MIC of meropenem against JLKP186. The calculated fractional inhibitory concentration index (FICI) was less than 0.5 (0 + 0.25/16, 0 + 0.5/16), indicating a strong synergistic interaction between phages and meropenem. However, the synergy between JLBP1002 and meropenem was observed to be unstable.

## JLBP1001 and JLBP1002 significantly improve survival in a mouse IAI model

Without intervention, intraperitoneal injection of $2.4 \times 10^8$ colony-forming units (CFUs)/mouse resulted in 100% mortality within 48 h in the infection group, whereas the survival rate of mice treated with phages 1 h post-infection was improved to varying degrees (Fig. 3A). Among the three treatment groups, the JLBP1001 treatment group had the most pronounced beneficial effect, with a 100% survival rate of mice across all MOIs. In contrast, the JLBP1002 treatment group showed a markedly reduced efficacy, with only a 40% survival rate at an MOI of 10. The phage cocktail treatment group, which consisted of an equal-volume mixture of both JLBP1001 and JLBP1002 administered at the same total dose as the monotherapies, displayed a slightly reduced therapeutic effect, achieving a 90% survival rate.

To further assess the therapeutic effect of phage therapy in the mice IAI model, the bacterial load in various organs was quantified 12 h post-infection (Fig. 3C through E). We found that while the bacterial burden in the liver, spleen, lungs, and kidneys of the three treatment groups exhibited only slight differences under all of the tested MOIs,

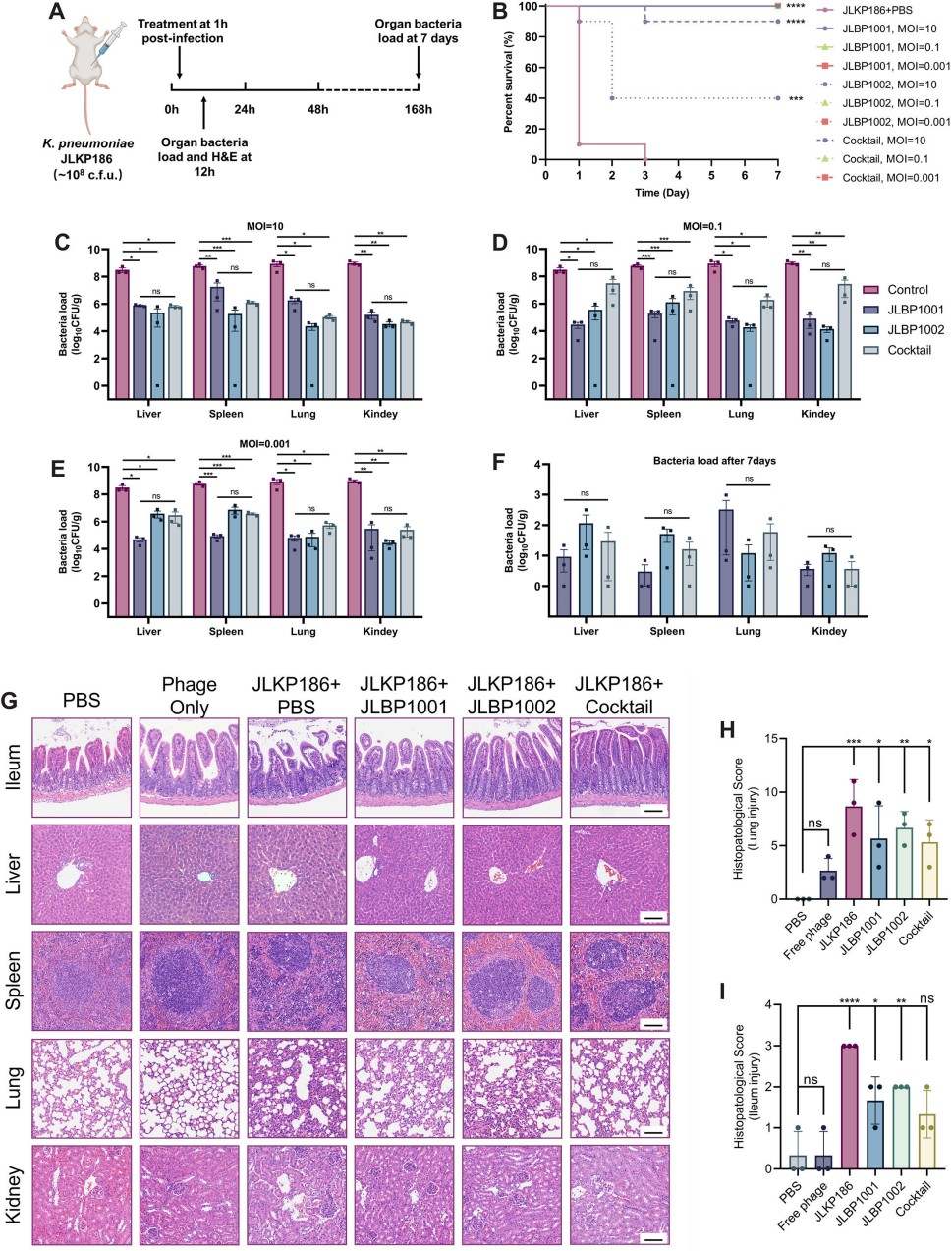

**FIG 3** The *in vivo* therapeutic effects of JLBP1001 and JLBPP1002. (A) The mouse model of IAI. (B) Survival curve of CD1 mice using JLKP186 and post-1 h treated with JLBP1001 and JLBP1002 (*n* = 10 per strain). The phage cocktail treatment group consisted of an equal-volume mixture of JLBP1001 and JLBP1002 administered at the same total dose as the monotherapies. Statistical analysis was performed using the log-rank (Mantel-Cox) test (***: *P* < 0.001, ****: *P* < 0.0001). (C–E) Bacterial load in the liver, spleen, lung, and kidney of mice (*n* = 3) infected with JLKP186 and treated with phages, measured at 12 h post-treatment. (C) MOI = 10; (D) MOI = 0.1; (E) MOI = 0.001. Each plot represents an individual mouse, and error bars indicate mean ± SEM. Data were analyzed by one-way ANOVA and Tukey's multiple comparisons test (*: *P* < 0.05, **: *P* < 0.01, ***: *P* < 0.001). (F) Bacterial load in the liver, spleen, lung, and kidney of mice (*n* = 3) infected with JLKP186 and treated with phages, measured at day 7 post-treatment. Each symbol represents a mouse, and error bars indicate mean ± SEM. Data were analyzed by one-way ANOVA and Tukey's multiple comparisons test. (G) Mice were randomly divided into six groups: phosphate-buffered saline (PBS) group (negative control), bacteria-infected group (positive control), phage-only group (free phage), and phage-treated group (JLBP1001, JLBP1002, and Cocktail, MOI = 10). Scale bar = 100 μm. (H–I) Histopathological scores of lung and ileum from CD1 mice. Histopathological scores (*n* = 3 biologically independent experiments) are shown on the right. Data analyses were performed using one-way ANOVA and Dunnett's multiple comparisons test (*: *P* < 0.05, **: *P* < 0.01, ***: *P* < 0.001, ****: *P* < 0.0001).

the bacterial burden in these organs from mice in the treatment groups was significantly lower compared to those in the infection group. In all the treatment groups, the highest bacterial load was approximately 963 CFUs/g in the lung tissue from mice that survived for 7 days post-infection, significantly lower than that in the 12 h treatment group (Fig. 3F). Furthermore, both JLBP1001 and JLBP1002 decrease the bacterial load from the initial $10^8$ CFUs/g to around $10^3$ CFUs/g within 7 days, further demonstrating the feasibility of phage therapy in the intraperitoneal infection model.

We then performed hematoxylin and eosin (H&E) staining on the ileum, liver, spleen, lungs, and kidneys of mice from each experimental group (Fig. 3G through I). Compared to the control group, the infection group demonstrated a high histopathological score and significant infiltration of inflammatory cells, cellular edema, and cell necrosis in various tissues. In contrast, the three treatment groups showed varying degrees of reduced tissue damage, indicating that phage therapy alleviated inflammation caused by bacterial infection.

## Phage-resistant strains exhibit significant fitness trade-offs

Although phages can rapidly lyse bacteria, the emergence of phage-resistant bacterial strains can lead to the failure of phage therapy. To determine whether phage resistance emerged during treatment, we isolated 72 *K. pneumoniae* colonies from surviving mice in different treatment groups. The phage sensitivity of each isolate toward JLBP1001 and JLBP1002 was assessed using the cross-streak assay (Fig. S4).

Finally, 21 strains of bacteria with complete or partial phage resistance were identified and named RM01 to RM21 (resistant mutant) (Table 1). To assess the stability of phage resistance in these RM strains, continuous subculturing was conducted for 10 days, approximately 300 generations, and the phage resistance of each generation was evaluated. The results demonstrated that the resistance stability of the RM strains was relatively high, and no changes in resistance were observed during the 10 days of subculturing (Fig. S5).

To evaluate the virulence of RM strains, the mouse intraperitoneal infection model was established by injecting $6 \times 10^7$ CFUs/mL of a bacterial suspension into the peritoneal cavity, and the 7-day survival rate of the mice was observed. We found that most RM strains exhibited reduced virulence compared to the parent *K. pneumoniae* JLKP186 strain, but some resistant strains still exhibited increased virulence, such as RM03 and RM21 (Fig. 4A). We then performed capsule production, bacterial viscosity, and biofilm formation assays on the 21 RM strains. The results revealed that most RM strains displayed reduced capsule production, bacterial viscosity levels, and increased biofilm expression (Fig. 4B through D). However, among them, both RM03 and RM21 not only tested positive in the string test but also did not exhibit reduced bacterial viscosity (Fig. 1C). The phage adsorption efficiency assay showed that the phage adsorption rate of all 21 RM strains decreased to varying degrees compared with the parent strain JLKP186 (Fig. 4E and F).

## Key loci of phage-resistant *K. pneumoniae in vivo* occurred in the CPS gene cluster

To further reveal the mechanism by which resistance to phages was developed by the RM strains, WGS was conducted on the 21 RM strains. Several single nucleotide polymorphism (SNP) differential genes were obtained after analysis with Snippy (https://github.com/tseemann/snippy) using the parent *K. pneumoniae* strain, JLKP186 (Table 1). The results indicated that 76.19% (16/21) of the strains have mutations in the CPS gene cluster, primarily including genes *wbaP*, *wzc*, and *wzy* (Fig. 4G). In the glycosyltransferase-encoding *wbaP*, which was involved in CPS biosynthesis, two mutations were identified (frameshift_variant and missense_variant). In the *wzc*, which encodes a tyrosine-protein kinase, four mutations were identified: frameshift_variant, stop_gained, frameshift_variant, and frameshift_variant. In the glycosylation-encoding *wzy*, two mutations were

**TABLE 1** Information on 21 RM strains was recovered from the phage treatment group in 7 days[a]

| Strains | Resistant isolate | Source | Mutation | Type | Effect | Putative function |
|---|---|---|---|---|---|---|
| RM01 | JLBP1001 | Spleen | *putA* | snp | 1894T>C | Proline metabolism |
| RM02 | JLBP1001 | Spleen | *putA* | snp | 1894T>C | Proline metabolism |
| RM03 | JLBP1001 | Spleen | – | | | |
| RM04 | JLBP1001 | Spleen | *putA* | snp | 1894T>C | Proline metabolism |
| | | | *wbaP* | del | 910_911delAT | CPS gene cluster |
| RM05 | JLBP1002 | Spleen | *wbaP* | del | 910_911delAT | CPS gene cluster |
| RM06 | JLBP1002 | Spleen | – | | | |
| RM07 | JLBP1002 | Spleen | *wzc* | ins | 1798dupG | CPS gene cluster |
| RM08 | JLBP1002 | Spleen | *wbaP* | del | 910_911delAT | CPS gene cluster |
| RM09 | JLBP1002 | Spleen | *putA* | snp | 1894T>C | Proline metabolism |
| | | | *wzc* | snp | c.1180C>T | CPS gene cluster |
| RM10 | JLBP1002 | Spleen | *wbaP* | snp | c.1003A>G | CPS gene cluster |
| RM11 | JLBP1002 | Spleen | *wzc* | del | 27_30delAACA | CPS gene cluster |
| RM12 | JLBP1001+1002 | Spleen | *putA* | snp | 1894T>C | Proline metabolism |
| RM13 | JLBP1001+1002 | Spleen | *wbaP* | del | 910_911delAT | CPS gene cluster |
| RM14 | JLBP1001+1002 | Spleen | *wzy* | del | 651_652delAT | CPS gene cluster |
| RM15 | JLBP1001+1002 | Spleen | *wbaP* | del | 910_911delAT | CPS gene cluster |
| RM16 | JLBP1001+1002 | Spleen | *wbaP* | del | 910_911delAT | CPS gene cluster |
| RM17 | JLBP1001+1002 | Spleen | *wzy* | del | 651_652delAT | CPS gene cluster |
| RM18 | JLBP1001 | Lung | *wzy* | ins | 262dupG | CPS gene cluster |
| RM19 | JLBP1001 | Lung | *wzy* | ins | 262dupG | CPS gene cluster |
| RM20 | JLBP1001+1002 | Lung | *wzy* | ins | 262dupG | CPS gene cluster |
| RM21 | JLBP1001+1002 | Lung | *wzc* | ins | 27_30dupAACA | CPS gene cluster |

[a]"–" represents unidentified SNPs; the "RM" series mutants are isolated from the surviving mice in the treatment group.

found: frameshift_variant and frameshift_variant. Additionally, mutations in the *putA* were found in RM01, RM02, RM04, RM09, and RM12 strains, and the *putA* is involved in the metabolism of proline to glutamate. In *Bacillus subtilis*, knockout of the *putA* has been shown to confer phage resistance (14).

## The CPS gene cluster and transcriptional regulators are downregulated in phage-resistant strains

To investigate the mechanisms underlying phage resistance in RM01 and RM12, we conducted metabolomic analysis (Fig. S6A through C). According to the metabolomic analysis, unsupervised principal component analysis (PCA) revealed that the metabolite environment of RM01 and RM12 was significantly altered as compared to JLKP186. The volcano plot showed differentially expressed metabolites, while Kyoto Encyclopedia of Genes and Genomes (KEGG) enrichment analysis found that the biosynthesis of cofactors, pantothenate and CoA biosynthesis, and alanine, aspartate, and glutamate metabolism were the most affected pathways. However, the relative contents of L-proline and L-glutamic acid were reduced in strains RM01 and RM12 (Fig. 5A).

To further investigate the specific mechanisms by which phage resistance is conferred to RM01 and RM12, transcriptomic analysis was conducted on JLKP186, RM01, and RM12. In the transcriptomic analysis, JLKP186, RM01, and RM12 jointly expressed 4,969 genes. Compared to the control group, the resistant strains upregulated 192 and downregulated 254 differentially expressed genes (DEGs) (Fig. 5C). A total of 458 genes were included in the Gene Ontology (GO) enrichment analysis, with 252 upregulated genes (55.02%) and 206 downregulated genes (44.98%) (Fig. 5D). The upregulated DEGs were primarily enriched in cellular functions and biological processes related to cellular respiration and energy production and transfer. The downregulated DEGs were mainly enriched in biological processes, such as xenobiotic catabolic process, secondary

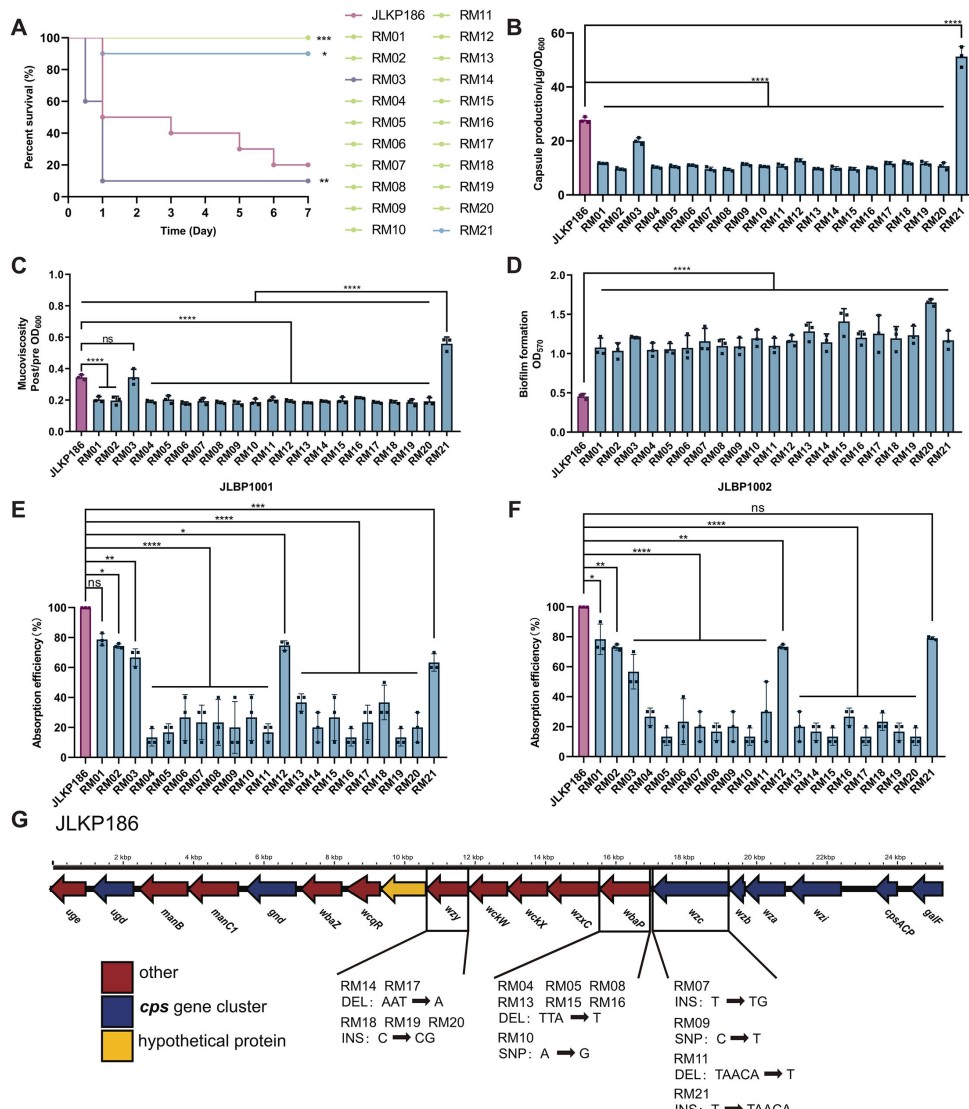

**FIG 4** The phenotype of phage-resistant mutants. (A) Survival curves of CD1 mice infected with RM strains and JLKP186 (*n* = 10 per strain). Data analysis was performed using the log-rank (Mantel-Cox) test (*: *P* < 0.05, **: *P* < 0.01, ***: *P* < 0.001). (B) Capsule production, (C) mucoviscosity, (D) biofilm formation, and (E–F) absorption efficiency of RM strains and JLKP186. Bars indicate mean ± SD. Each plot represents the mean of three technical replicates from one independent experiment. Three independent experiments were performed. Statistical analyses were performed using one-way ANOVA and Dunnett's multiple comparisons test (*: *P* < 0.0001). (G) CPS gene cluster of JLKP186. Variant types are labeled as SNP (single nucleotide polymorphism), DEL (deletion), and INS (insertion).

metabolic process, and glutamine family amino acid catabolic process (Fig. S6D through F).

The KEGG enrichment analysis included a total of 269 genes, of which 117 genes (43.49%) were upregulated, and 152 genes (56.51%) were downregulated (Fig. 5E). The KEGG enrichment analysis based on upregulated DEGs demonstrated that the DEGs were primarily concentrated in pathways, such as the citrate cycle (TCA cycle), carbon fixation pathways in prokaryotes, and oxidative phosphorylation. The downregulated DEGs were focused on some amino acid metabolic pathways (Fig. S6G through I).

Interestingly, despite the absence of mutations in genes encoding CPS and its transcriptional regulators (*rmpA*, *rmpA2*, and *rcsA*), both RM01 and RM12 displayed significant transcriptional downregulation of these genes. In contrast, the proline

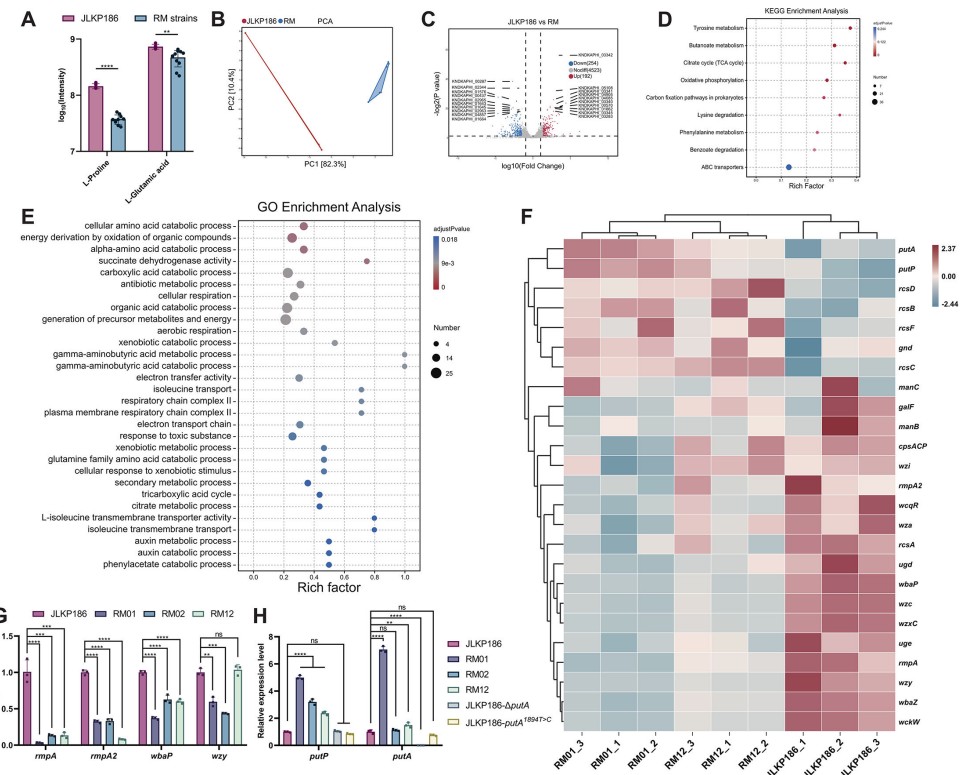

**FIG 5** Transcriptome analysis of JLKP186 and RM strains. (A) The intensity of L-proline and L-glutamic acid in the metabolomic analysis. (B) PCA analysis of the transcriptome. (C) Volcano plot showing upregulated (red) and downregulated (blue) DEGs analysis of JLKP186 and RM strains. (D) KEGG enrichment analysis (rich factor). (E) GO enrichment analysis (rich factor). (F) Heat map representing the expression gene related to CPS cluster and transcriptional regulator in JLKP186 and RM strains. Blue to red colors correspond to low to high abundance. (G–H) Expression levels of selected genes by RNA sequencing. The expression levels of genes were identified by qRT-PCR. Experiments were carried out in triplicate with three independent RNA samples. Student's *t*-test, as implemented in GraphPad Prism software 8, was used to analyze the statistical significance of the differences (*: *P* < 0.0001).

metabolism–related genes *putA* and *putP* showed a trend of increased transcription in RM01 and RM12, as revealed by the transcriptome analysis (Fig. 5F). To validate the RNA-sequencing (RNA-seq) results, we selected six genes, including *rmpA*, *rmpA2*, *wbaP*, *wzy*, *putA*, and *putP*, for quantitative PCR (qPCR) verification in RM01 and RM12. The results indicated that the trends were consistent with the transcriptomic measurements (Fig. 5G and H).

## Functional exploration of *putA* in phage-resistant *K. pneumoniae*

To further explore the specific role of the *putA* gene in the phage resistance of *K. pneumoniae*, we constructed a series of genetically engineered strains based on the JLKP186 background. These included the *putA* knockout strain (JLKP186-Δ*putA*), its complemented strain (JLKP186-Δ*putA*::pputA), and a complemented strain generated in the resistant mutant background (RM12-pputA). In addition, we successfully constructed a point-mutant strain (JLKP186-*putA*^1894T>C^) to mimic the naturally occurring mutation identified in the resistant strain RM12.

We first characterized the biological properties of these strains, including CPS production, mucoviscosity, biofilm formation, and absorption efficiency (Fig. 6). The results demonstrated that neither deletion, complementation, nor point mutation of *putA* caused significant changes in these phenotypic traits when compared with the parental JLKP186 strain or the RM12, suggesting that *putA* alone is not a major determinant of these bacterial biological characteristics.

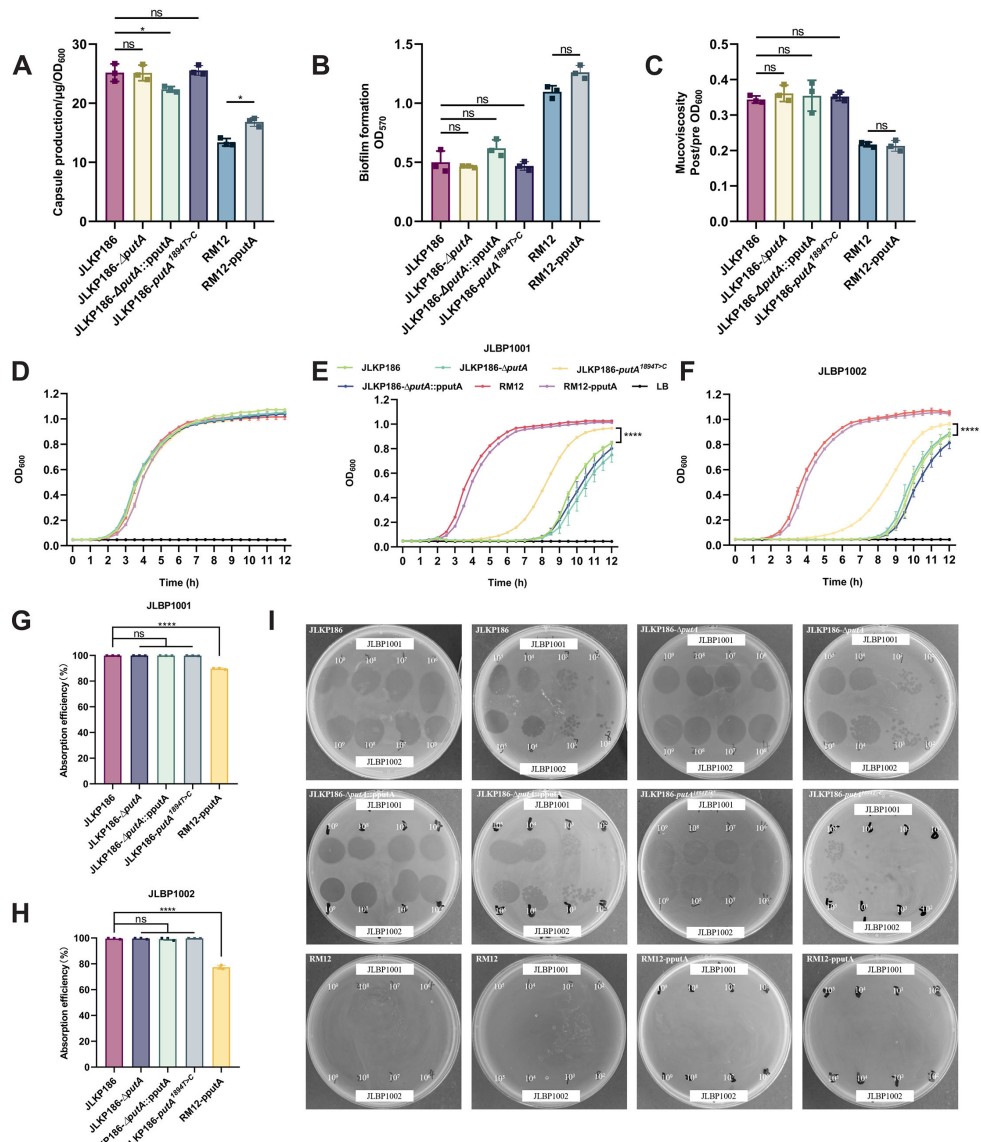

FIG 6 The phenotype of *putA* knockout and complement strains. (A) Capsule production, (B) mucoviscosity, and (C) biofilm formation of these strains and JLKP186. (D–F) Growth curve of these strains in LB medium. The MOI of phages and strains is 0.0001. Each plot represents the mean of three technical duplications. Statistical analyses were performed using two-way ANOVA. (E) [Interaction: $F (24, 100) = 221.3$, $P < 0.0001$; row factor: $F (24, 100) = 6,399$, $P < 0.0001$; column factor: $F (1, 100) = 3,243$, $P < 0.0001$]. (F) [Interaction: $F (24, 100) = 22.35$, $P < 0.0001$; row factor: $F (24, 100) = 846.2$, $P < 0.0001$; column factor: $F (1, 100) = 284.2$, $P < 0.0001$]. (G and H) Absorption assay of RM strains. (I) Spot assays of putA knockout and complement strains. The phage concentrations were diluted from $10^9$ to $10^2$ PFUs/mL. Error bars indicate mean ± SD. Each plot represents the mean of three biological replicates for each group. Statistical analyses were performed using one-way ANOVA and Dunnett's multiple comparisons test (*: $P < 0.0001$).

We monitored the bacterial growth of all engineered strains in liquid medium without phage exposure and found that their growth kinetics were highly similar, with no marked differences among these strains (Fig. 6D). We then assessed the *in vitro* bactericidal activity of the two phages in liquid culture (Fig. 6E and F). Growth curve analysis showed that JLKP186-Δ*putA* and JLKP186-Δ*putA*::pputA exhibited phage susceptibility similar to the wild type, with nearly overlapping lysis curves. However, the JLKP186-*putA*^1894T>C strain exhibited an earlier recovery of growth under phage challenge compared to the wild-type (WT) strain. By contrast, RM12 and RM12-pputA maintained robust growth

under phage challenge, consistent with their resistant phenotype. The absorption efficiency assay demonstrated that the adsorption rate of these knockout, complementation, and point mutation strains was almost the same as that of JLKP186 (Fig. 6H and I). Spot assays further supported these findings (Fig. 6J). *putA* deletion and complementation did not alter phage sensitivity, whereas the JLKP186-*putA*$^{1894T>C}$ point mutant displayed reduced plaque formation efficiency. Clear plaques were observed only at higher phage titers, while plaques became progressively faint and discontinuous with decreasing titers, eventually disappearing.

## DISCUSSION

In this study, we utilized a clinical CR-hvKP strain of the KL57 serotype to isolate and characterize two lytic phages. The therapeutic efficacy of both phages in an IAI model was confirmed through comprehensive biological profiling. Phage resistance has been extensively studied under *in vitro* conditions; however, the evolutionary trajectories and mechanisms of phage resistance that emerge within host environments remain poorly understood. In this study, by integrating WGS, metabolomic analysis, and transcriptomic analysis, we investigated the phage resistance mechanisms of *K. pneumoniae* isolates that evolved under phage selection pressure in a murine IAI model. This *in vivo*–derived data set enabled us to explore bacterial adaptive strategies shaped by host-associated conditions.

Findings from a prior study indicated that the CPS structure of bacteria is one of the key receptors for phage recognition in *K. pneumoniae*, and various degrees of alteration in CPS structure can lead to the emergence of phage resistance (15–17). In addition, mutations in other genes involved in CPS synthesis, including *wbaZ*, *mshA*, *wcaJ*, *wcal*, and *wza*, have also been shown to confer phage resistance (18–22). In the present study, most of the *in vivo* RM strains isolated obtained phage resistance through mutations at different sites of the CPS gene cluster, including *wbaP* (7/21), *wzc* (4/21), and *wzy* (5/21). Thus, the development of phage resistance mutations in *K. pneumoniae* isolated from surviving mice is not limited to a specific location within the CPS gene cluster but is rather distributed across various genes.

A previous study confirmed the efficacy of phage therapy for multidrug-resistant *K. pneumoniae* infection in animal models (23). Here, by constructing the mouse model of IAI with CR-hvKP, we demonstrated the efficacy of phage therapy for CR-hvKP infection. Both monotherapy and cocktail therapy significantly increased the 7-day survival rate of mice and reduced the bacterial burden in organs. Notably, the decreased efficacy of JLBP1002 monotherapy, along with its detrimental effect on cocktail treatment, suggests that JLBP1002 may not be suitable for high-MOI *in vivo* applications. The reduced therapeutic effect of JLBP1002 at high MOI may be attributed to the rapid bacterial lysis induced by the high phage concentration, leading to massive bacterial death and a transient endotoxemia that accelerated mouse mortality (24, 25). Although the bacteria were not totally eliminated after 7 days, the virulence of the surviving bacteria in the mice was considerably reduced. At the same time, *in vitro* experiments confirmed that the mucoviscosity of almost all phage-resistant strains decreased, and the production of CPS, as well as biofilm formation, decreased, confirming the adaptability cost of phage-resistant bacteria in prior investigations (17, 26–29). However, during the course of the experiment, not all phage-resistant strains exhibited a trend of decreasing virulence, which differed significantly from earlier studies (30).

The hypermucoviscous phenotype of RM03 and RM21 made us realize that their virulence may not have been reduced, validated in a mouse IAI model. Moreover, no significant mutations were identified in RM03 using Snippy. However, further phage resistance testing via gradient dilution revealed that RM03 only displayed resistance to JLBP1002, rather than full resistance to both phages (Fig. S5). qPCR analysis revealed that, compared with the WT strain, RM03 exhibited a marked reduction in the expression of the regulators *rmpA* and *rmpA2*, accompanied by varying degrees of upregulation in the capsule biosynthesis genes *wbaP*, *wzy*, and *wzc* (Fig. S6J). These data collectively indicate

that RM03 likely represents a form of regulatory-driven adaptive resistance, in which capsule-associated pathways are transcriptionally compensated despite the reduced expression of *rmpA* and *rmpA2*. Such a mechanism may allow RM03 to mitigate phage adsorption while avoiding the typical fitness trade-offs observed in capsule-deficient resistant mutants, such as decreased mucoviscosity or attenuated virulence.

Through comparative genomic analysis, we identified a frameshift mutation in the *wzc* gene of RM21 (frameshift_variant c.27_30dupAACA, p.Pro11fs), which results in a premature stop codon at the 17th amino acid. Notably, several other isolates carrying early *wzc* truncations (e.g., RM07, RM09, and RM11) displayed the expected capsule-deficient and virulence-attenuated phenotypes. However, previous studies have demonstrated that mutations in *wcaJ* (also known as *wbaP*) and *wzc* lead to distinct phenotypic outcomes: *wcaJ* mutations typically abolish capsule synthesis, whereas *wzc* functions as a key regulator of capsule quantity and architecture (31). In addition, another study reported that mutations occurring in *wzc* could confer phage resistance while preserving the hypermucoviscous phenotype and partially retaining virulence (CR5 *wzc*$^{Y561D}$) (32). Therefore, it is plausible that the specific frameshift event in RM21 may modulate capsule structural organization rather than causing complete capsule loss, contributing to its retained virulence. These two RM strains with special phenotypes are worth analyzing further to delineate their mechanisms of phage resistance. Additionally, it is necessary to be alert to the possibility of hypervirulent mutants in clinical applications. The PAS test between JLBP1001, JLBP1002, and meropenem suggested combining phage and antibiotic therapy to reduce the likelihood of hypervirulent mutants emerging (33, 34).

In naturally evolved resistant isolates RM01, RM02, and RM12, we identified a *putA* point mutation (*putA*$^{1894T>C}$). To investigate its functional relevance, we constructed the *putA* knockout strain, its complemented strain, and a point-mutant strain. The deletion or restoration of *putA* did not alter phage susceptibility, indicating that loss of *putA* function is not sufficient to reshape the phage–bacteria interaction. However, the point-mutant strain exhibited a reduced phage sensitivity phenotype, characterized by earlier recovery during phage challenge in liquid culture and diminished plaque formation efficiency on solid medium, suggesting that the *putA*$^{1894T>C}$ influences the phage infection process.

The *putA* gene is involved in the metabolism of proline to glutamate (35). Previous research has reported that deletion of *putA* in *B. subtilis* reduced the time window during which phage SPP1 can establish productive infection (14). It suggests that *putA*-associated metabolic states can influence phage infection efficiency. Other research indicated that supplementation with exogenous proline can reshape bacterial metabolism and modulate the expression of outer membrane proteins such as OmpA, OmpK35, and OmpK36, ultimately increasing bacterial susceptibility to serum complement-mediated killing (36). Therefore, we speculate that the *putA*$^{1894T>C}$ mutation in RM strains may alter the bacterial metabolic state, thereby affecting the phage infection process and ultimately contributing to the development of phage resistance. Further work will be required to determine how this mutation affects intracellular metabolic flux and whether such changes directly influence the phage infection process.

Overall, our study confirmed the therapeutic efficacy of phages JLBP1001 and JLBP1002 in a mouse IAI model and identified a metabolically driven pathway associated with phage resistance *in vivo*. However, although genomic and transcriptomic analyses suggested candidate resistance-associated pathways, we did not functionally validate all of the putative determinants, and additional genetic or biochemical confirmation will be necessary. Second, the potential influence of the complex host immune environment on resistance evolution remains unclear, and phage–host–immune interactions warrant further investigation. Finally, several frequently mutated regions identified in resistant isolates remain uncharacterized, and future work should dissect their regulatory roles and contributions to phage adaptation. Despite these limitations, our findings provide a conceptual framework for understanding metabolic adaptation-mediated phage resistance and highlight directions for improving phage-based therapeutic strategies.

## MATERIALS AND METHODS

### Phage isolation and purification

Sewage samples were collected from the untreated wastewater processing station at Jinling Hospital in Nanjing, China, in 2023 to isolate phages. The sewage was centrifuged at 4,500 rpm at 4°C for 10 min, and the supernatant was filtered through the 0.22 µm microporous filter. Using the double-layer agar method, individual and clear plaques were obtained from the supernatant. To isolate individual phages, the plaques were further purified at least five times until uniform and clear plaques formed. Using JLKP186 as the host strain, we obtained two lytic bacteriophages, named JLBP1001 and JLBP1002.

### Host spectrum determination

We utilized double-layer agar assays and spotting tests to evaluate the host range of the isolated phages. Specifically, 10 µL of a solution enriched with phages (about $\sim10^9$ PFUs/mL) was dropped on a plate with a confluent lawn of the *K. pneumoniae* host strain in order to determine the lysis spectrum. The plate was incubated at 37°C for more than 4 h and then observed for the formation of plaques.

### Antimicrobial susceptibility test

The MIC of antibiotics was evaluated utilizing the broth dilution method in accordance with the Clinical and Laboratory Standards Institute (CLSI) (37). In addition, to eliminate the effect of growth rate on the analysis, the plate incubation time was extended to 48 h. Briefly, 100 µL of twofold diluted antibiotic solution was added to the wells of a 96-well microtiter plate, followed by 100 µL of overnight bacterial culture. After thorough mixing, the plates were incubated at 37°C for 48 h. Results were interpreted according to the guidelines provided in CLSI document M100 (38). *K. pneumoniae* ATCC 700603, *Escherichia coli* ATCC 29212, and *Pseudomonas aeruginosa* ATCC 27853 were used as quality control strains.

### WGS analysis

The genomic DNA of 5 mL of phage (about $10^{10}$ PFUs/mL) was extracted using the λPhage Genomic DNA Extraction Kit (Cat. #AB1141, Abigen, China). The genome was sequenced using the Illumina HiSeq system (Illumina, San Diego, CA, USA). Briefly, genomic DNA samples were fragmented by sonication to a size of 350 bp. The DNA fragments were then end-polished, A-tailed, and ligated with a full-length adapter for Illumina sequencing, followed by further PCR amplification. Subsequently, library quality was evaluated and quantified by qPCR. The qualified libraries were pooled and sequenced on an Illumina platform with the PE150 strategy, according to the effective library concentration and data amount required. PhageScope (39) was employed to identify the phage lifestyle. All strains were annotated with Prokka v 1.14.6 (40) and Pharokka v 1.7.2 (11).

### TEM

Morphological characterization of phages JLBP1001 and JLBP1002 was conducted utilizing TEM, beginning with the preparation of a high-titer phage enrichment solution, resulting in a measured titer above $10^{10}$ PFUs/mL. Approximately 20 µL of the enrichment was dropped onto a copper grid with carbon film for 3–5 min. Filter paper was then used to absorb the excess liquid. Then, 2% phosphotungstic acid was placed on the copper grid to stain for 1–2 min. Filter paper was used to absorb the excess liquid, and the slide was dried at 27°C. The copper grids were observed under TEM (Hitachi-HT7800), and images were taken.

## One-step growth curve assay

One-step growth curve methods were employed to assess the phage growth cycles and burst size. Each phage (1 mL) was mixed at an MOI of 0.1, shaken at 180 rpm, and incubated at 37°C for 5 min. The solution was then centrifuged at 4,500 rpm for 3 min, and the supernatant was discarded. The pellet was washed with PBS, resuspended in 10 mL of LB medium, and incubated for 120 min at 37°C and 180 rpm. The phage titer was then determined every 10 min using the double-layer agar method.

## Optimal MOI

The experimental procedure was conducted according to previously published methods (33, 41). To confirm that the optimal MOI produces the largest amount of progeny phages during bacterial lysis, the host strain JLKP186 was grown to logarithmic phase and adjusted to $1.5 \times 10^8$ CFUs/mL. Then, JLKP186 and the phages were mixed at different MOIs (100, 10, 1, 0.1, 0.01, 0.001, and 0.0001) for 10 min, after which the number of progeny phages was counted utilizing the double-layer agar method.

## pH and temperature stability

For temperature stability, a phage solution at $10^8$ PFUs/mL was incubated at 4, 16, 37, 50, and 60°C for 1 h, after which the phage titer was determined using the drop method. For pH stability, we prepared SM buffers with different pH levels (2–11) using NaOH or HCl. Then, 100 µL of the phage solution was mixed with 900 µL of the buffer at different pH levels and incubated at 37°C for 1 h. The phage titer was subsequently determined using the drop method.

## Biofilm formation assays

Static biofilm formation of *K. pneumoniae* was evaluated using crystal violet staining. To evaluate the biofilm formation of *K. pneumoniae*, we adjusted an overnight culture of *K. pneumoniae* to $10^8$ CFUs/mL. Then, 10 µL of the bacterial culture was added to a 96-well plate, followed by 190 µL of LB liquid medium. The plate was then incubated in a 37°C incubator for 48 h. The supernatant was then removed from the wells, and the wells were then washed three times with PBS, stained with crystal violet for 15 min, and then washed three times with PBS. We then added 200 µL of absolute ethanol to each well for 5 min to elute the crystal violet from the bacteria. The $OD_{570}$ was measured using a microplate reader using an endpoint method.

## Mucoviscosity assays

The bacterial strains were grown overnight in LB liquid medium, followed by centrifugation at $1,000 \times g$ at 4°C for 5 min. The $OD_{600}$ ratio of the supernatant after centrifugation compared to the initial culture broth ($OD_{supernatant}/OD_{culture\ broth}$) was determined.

## *In vitro* bacteriolytic characteristics of phages

To confirm the bacteriolytic characteristics of two isolated phages *in vitro*, the bacteriolytic curve of each phage was determined by mixing 100 µL of JLKP186 overnight culture with 100 µL of LB medium or phage enrichment solution at various MOIs. The OD values of each co-culture were measured at 30-min intervals for up to 12 h.

## Capsule production assays

To determine capsule production, we mixed 500 µL of *K. pneumoniae* overnight culture with 100 µL of citric acid and incubated the mixture at 50°C for 20 min. Then, the mixture was centrifuged at 10,000 rpm for 5 min. Next, 250 mL of the mixture supernatant was mixed with 1.2 mL of absolute ethanol, incubated at 4°C for 20 min, and centrifuged at the maximum speed for 5 min. After precipitation and drying, the pellet

was resuspended in 200 µL pure water + 1.2 mL tetrathiol and then heated at 100°C for 5 min. Lastly, 20 µL of triphenyl NaOH was added to the mixture, incubated for 5 min, and then $OD_{520}$ and $OD_{600}$ were determined. The CPS content was calculated using a glucuronic acid standard curve and normalized as µg/$OD_{600}$.

## Absorption efficiency assay

One milliliter of overnight bacterial cultures was adjusted to $10^8$ CFUs/mL, and 10 µL of the diluted phage suspension was mixed at an MOI of 0.01. The mixture was incubated at 37°C and shaken at 180 rpm for 10 min to allow phage adsorption. Subsequently, the culture was centrifuged at 12,000 × $g$ for 5 min at 4°C, and the supernatant was collected and passed through a 0.22 µm sterile membrane filter to remove bacterial cells. The titer of unadsorbed phages remaining in the filtrate was determined by serial dilution followed by the double-layer agar plaque assay. Absorption efficiency = 1 − (residual phage titer/initial phage titer).

## Phage-antibiotic synergistic effects

According to the CLSI 2022 standards and previous studies (37, 42), meropenem and imipenem were diluted in the cation-adjusted Mueller-Hinton broth (CAMHB) medium. The FICI involved adding various concentrations of phages (50 µL, 1:10) and antibiotics (100 µL, 1:2) in a 96-well plate, along with 50 µL of a bacterial suspension ($10^6$ CFUs/well). CAMHB medium was then used to bring the volume up to 200 µL. The mixture was then incubated at 37°C for 16–18 h, and the MIC was measured by $OD_{600}$.

## Mouse IAI model

All strains were cultured for 8 h in 5 mL of LB medium. The bacterial suspension was then centrifuged at 3,900 rpm for 10 min, followed by two washes with PBS. The bacterial suspension was adjusted to $10^9$ CFUs/mL. Mice (CD1, 4 weeks old) were randomly divided into groups of 10, and 200 µL of the bacterial suspension was injected into the lower left abdomen. The treatment group received phage therapy with 200 µL but various MOIs 1 h after bacterial injection. Mice were observed for 7 days.

## Bacterial load measurement and histopathological examination

To determine the bacterial burden in the mice during phage therapy, three mice from each group were randomly selected 12 h after phage inoculation to collect organ tissue. We collected 0.05 g (± 0.005) of liver, spleen, lung, and kidney tissue, added 1 mL PBS, and then homogenized. The tissue homogenate was diluted 10-fold to $10^{-1}$, $10^{-2}$, and $10^{-3}$ and spread on LB agar containing 100 µg/mL ampicillin for colony counting. To observe the histopathological changes, part of the tissue was also fixed in 4% paraformaldehyde, followed by dehydration, paraffin embedding, and staining with H&E.

## Spot assay and cross-streak assay

One hundred microliters of a bacterial suspension was spread on an LB agar plate, and then 10 µL of various concentrations of phage suspension ($10^9$–$10^2$ PFUs/mL) was dropped onto the agar covered by bacterial suspension. Then, incubated at 37°C for 4–6 h, the clear plaque caused by phage on the lawn indicates sensitivity.

Ten microliters of the diluted phage suspension was streaked along the central line of an agar plate. After the surface was air-dried, bacterial cultures were sequentially streaked across the phage line in a perpendicular direction. The plates were then air-dried again and incubated at 37°C for 8 h.

## Cloning and complementation experiments

The *putA* deletion mutant of *K. pneumoniae* JLKP186 was generated using a CRISPR/Cas9-based genome editing system as previously described (43). Briefly, the upstream

and downstream homologous arms of *putA* were PCR-amplified and assembled into pUC19 to produce the repair template. Cas9-expressing JLKP186 cells were co-transformed with the repair fragment and the *putA*-targeting sgRNA plasmid, and mutants were selected on kanamycin and spectinomycin plates. Candidate clones were verified by PCR across the deletion junction and confirmed by Sanger sequencing. A complementation plasmid carrying WT *putA* was subsequently constructed using the pSCC19 vector.

The *putA* point-mutant strain was constructed using an allelic exchange strategy based on a sacB-containing suicide plasmid. Briefly, the *putA* fragment harboring the desired point mutation was amplified from strain RM12 genomic DNA using primers listed in Table S3 and cloned into the suicide vector pAprsacB via seamless cloning. The resulting plasmid was transformed into *E. coli* S17 λpir and verified by colony PCR and Sanger sequencing. The verified plasmid was then introduced into the JLKP186 strain by conjugation. Single-crossover recombinants were selected on LB agar plates containing appropriate antibiotics. Double-crossover mutants were subsequently obtained by counter-selection on LB plates supplemented with 10% (wt/vol) sucrose. Candidate colonies were screened by PCR and confirmed by Sanger sequencing.

## Metabolomic analysis

Preparation of bacterial samples and LC-MS/MS analysis were performed as described previously (44). Briefly, 100 μL of overnight *K. pneumoniae* culture was mixed with 400 μL of precooled methanol:acetonitrile (1:1, vol/vol), vortexed for 30 s, frozen at −20°C for 30 min, and centrifuged at 12,000 rpm for 10 min at 4°C. The supernatant was vacuum-dried and resuspended in 150 μL of 50% methanol containing 5 ppm 2-chlorophenylalanine, followed by centrifugation and filtration through a 0.22 μm membrane. A pooled QC sample was prepared by combining aliquots of each sample to monitor analytical stability.

Multivariate statistical analyses, including PCA, PLS-DA, and OPLS-DA, were performed using the R package *ropls*. Differential metabolites were identified based on VIP > 1 and $P < 0.05$, and their fold changes (FCs) were calculated. KEGG pathway enrichment analysis was subsequently conducted to elucidate enriched metabolic pathways.

## Transcriptomic analysis

Total RNA was isolated using the Trizol Reagent (Invitrogen Life Technologies). Quality and integrity were measured using a NanoDrop spectrophotometer (Thermo Scientific) and Bioanalyzer 2100 system (Agilent). A Zymo-Seq RiboFree Total RNA Library Kit was utilized to remove rRNA from the total RNA. The sequencing library was then sequenced on a NovaSeq 6000 platform (Illumina) by Shanghai Personal Biotechnology Cp. Ltd. Using the fragments per kilobase of transcript per million mapped reads values to identify $P$ values ≤ 0.05 and FC values ≥ 2 (log2 FC ≥ 1 or log2 FC ≤ −1). Values were aligned to the KEGG and GO databases for annotation, and DEGs were identified utilizing DESeq2.

## RT-qPCR analysis

An overnight bacterial culture was obtained as described previously, and the cells were washed for precipitation twice using PBS. Total RNA was extracted using the RNAprep Pure Cell/Bacteria Kit (Tiangen, DP430). After the total RNA was obtained, reverse transcription was conducted using HiScript III RT SuperMix for qPCR (+gDNA wiper). RT-qPCR analysis was performed on a 7500 Fast Real-Time PCR System (Applied Biosystem, CA, USA) with Taq Pro Universal SYBR QPCR Master Mix (Vazyme) and corresponding gene primers (Table S4). A two-step PCR amplification standard procedure was used for thermal cycling: 40 cycles of 95°C for 30 s, 95°C for 10 s, and 60°C for 30 s. The $2^{-\Delta\Delta Ct}$ method was used to calculate the FC in mRNA expression relative to the reference gene (16S rRNA).

## Statistical analysis

Statistical analyses were performed using GraphPad Prism 8.0.1. All data were expressed as the mean ± standard deviation (SD) or standard error of the mean (SEM). For the bacterial load, $P$ values were calculated utilizing a one-way analysis of variance (ANOVA) and Tukey's multiple comparisons test. For the histopathological scores, biofilm assay, mucoviscosity assays, and capsule production assays, statistical significances were calculated by a one-way ANOVA and Dunnett's multiple comparisons test. The survival curves were analyzed by a log-rank (Mantel-Cox) test, and the qPCR expression levels were analyzed via Student's $t$-test. The levels of statistical significance were set as follows: * $P < 0.05$, ** $P < 0.01$, *** $P < 0.001$, and **** $P < 0.0001$.

## ACKNOWLEDGMENTS

We are very grateful to the laboratory members of the Institute of General Surgery for their assistance in the experiment.

L.D. and J.L. contributed equally to this study and share first authorship. L.D. and J.L. performed the experiments and wrote the manuscript. W.W., L.X., M.Q., and S.Y. established the mouse IAI model. J.W., W.W., S.T., Z.Z., and M.W. collected and analyzed the results of bacteria load experiment. Y.Z. reviewed and revised this manuscript. X.W. and J.R. jointly supervised this work. All of these authors take full responsibility for the content of the manuscript and have approved its submission.

## AUTHOR AFFILIATIONS

[1]Jinling Clinical Medical College, Nanjing University of Chinese Medicine, Nanjing, China

[2]Research Institute of General Surgery, Jinling Hospital, The Affiliated Hospital of Medical School, Nanjing University, Nanjing, China

[3]Research Institute of General Surgery, Jinling Hospital, The Affiliated Hospital of Medical School, Nanjing Medical University, Nanjing, China

[4]Research Institute of General Surgery, Jinling Hospital, School of Medicine, Southeast University, Nanjing, China

[5]Clinical Translational Research Center for Surgical Infection and Immunity of Nanjing Medical University, Nanjing, China

## AUTHOR ORCIDs

Liuqing Dou http://orcid.org/0009-0005-8481-5999
Jiayang Li http://orcid.org/0009-0006-1878-9427
Xiuwen Wu http://orcid.org/0000-0002-3813-3867
Jianan Ren http://orcid.org/0000-0002-4697-4762

## FUNDING

| Funder | Grant(s) | Author(s) |
| --- | --- | --- |
| Navigation Project of Clinical Research | 22LCYY-LH4 | Jianan Ren |
| Jiangsu Provincial Medical Innovation Center | CXZX202217 | Jianan Ren |
| Key Research and Development Program of Jiangsu Province | BE2022823 | Jianan Ren |

## AUTHOR CONTRIBUTIONS

Liuqing Dou, Data curation, Investigation, Methodology, Software, Writing – original draft | Jiayang Li, Data curation, Investigation, Methodology, Software, Writing – original draft | Wenqi Wu, Data curation, Investigation, Methodology, Software | Li Xu, Data curation, Investigation, Methodology, Software | Mingjie Qiu, Data curation, Investigation, Methodology | Shuanghong Yang, Data curation, Methodology, Software | Jiajie Wang, Data curation, Software | Sai Tian, Data curation | Zhitao Zhou, Software |

Meilin Wu, Data curation, Software | Yun Zhao, Conceptualization, Supervision | Xiuwen Wu, Project administration, Supervision, Writing – original draft | Jianan Ren, Funding acquisition, Project administration, Supervision, Writing – review and editing

## DATA AVAILABILITY

The genome information of phages JLBP1001 and JLBP1002 has been submitted to GenBank under accession numbers PP973870.1 and PP994489.1, respectively. All the *K. pneumoniae* genomic sequences and raw data of RNA-seq have been published in the National Center for Biotechnology Information database (PRJNA1256399). Metabolomics data have been uploaded to MetaboLights (45) repository with the study identifier MTBLS12480.

## ETHICS APPROVAL

Ethical approval for this study was granted by the Ethical Committee of Jinling Hospital (approval No. 2024DZKY-001-01 and DZYJSKT520206160002).

## ADDITIONAL FILES

The following material is available online.

### Supplemental Material

**Fig. S1 (mSystems01476-25-s0001.tif).** Workflow of isolation and characterization of mutants.
**Fig. S2 (mSystems01476-25-s0002.tif).** Biological characteristics of JLBP1001 and JLBP1002.
**Fig. S3 (mSystems01476-25-s0003.tif).** Genome information for JLBP1001 and JLBP1002.
**Fig. S4 (mSystems01476-25-s0004.tif).** Sensitivity of isolated strains to phage infection.
**Fig. S5 (mSystems01476-25-s0005.tif).** Phage sensitivity assay of RM strains.
**Fig. S6 (mSystems01476-25-s0006.tif).** Metabolomics and transcriptome analysis of JLKP186 and RM strains.
**Supplemental Material (mSystems01476-25-s0007.xlsx).** Strain information.
**Supplemental Tables (mSystems01476-25-s0008.docx).** Tables S1 to S4.

### Open Peer Review

**PEER REVIEW HISTORY (review-history.pdf).** An accounting of the reviewer comments and feedback.

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
