## [Reviewer comments · mSystems]

Investigation of the therapeutic efficacy and resistance mechanisms of lytic phages targeting ST218 KL57 CR-hvKP

Liuqing Dou, Jiayang Li, Wenqi Wu, Li Xu, Mingjie Qiu, Shuanghong Yang, Jiajie Wang, Sai Tian, Zhitao Zhou, Meilin Wu, Yun Zhao, Xiuwen Wu, and Jianan Ren

Corresponding Author(s): Jianan Ren, Jinling Clinical Medical College, Nanjing University of Chinese Medicine

Review Timeline:

Submission Date:	November 15, 2025
Editorial Decision:	December 8, 2025
Revision Received:	December 15, 2025
Accepted:	December 18, 2025

Editor: Dustin Hancks

Reviewer(s): Disclosure of reviewer identity is with reference to reviewer comments included in decision letter(s). The following individuals involved in review of your submission have agreed to reveal their identity: Ella R. Rotman (Reviewer #2)

Transaction Report:

DOI: <https://doi.org/10.1128/msystems.01476-25>

Re: mSystems01476-25 (**Investigation of the therapeutic efficacy and resistance mechanisms of lytic phages targeting ST218 KL57 CR-hvKP**)

Dear Prof. Jianan Ren:

The reviewers acknowledge your revisions have improved the overall quality of the manuscript. Reviewer 2 has noted some very minor outstanding points related to the text (see attached). Along with a few noted typos, please add some more clarification to the text on how you made the mutants present in the new figure.

Revision Guidelines

Sincerely,
Dustin Hancks
Editor
mSystems

Reviewer #1 (Comments for the Author):

The authors have now reasonably addressed my prior concerns. With this, and the corresponding effort to also cover the points raised by other reviewers, the manuscript is now ready for the Editor to consider publication, in my opinion. I have only a very

minor comment on a previous point (Comment 2 of Reviewer #1) which I do not need to review in an additional round:

- The authors make their case in their answer, but I now believe that Figure 3B should be tuned to be able to distinguish the many different conditions. An option could be to add a supplementary figure in which the survival curves of the two phages and the cocktail are separated in different panels. It would just be for the sake of clarity.

Reviewer #2 (Comments for the Author):

I commend the authors for taking the time and effort to address the concerns I had in the original draft, particularly the work for Figure 6. In general, they have taken care of all the issues brought up and clarified things, resulting in an improved manuscript. I find the changes to be satisfactory, with only minor comments attached.

I commend the authors for taking the time and effort to address the concerns I had in the original draft, particularly the work for Figure 6. In general, they have taken care of all the issues brought up and clarified things, resulting in an improved manuscript. I find the changes to be satisfactory with only minor comments below.

Minor comments:

Line 284: “we attempted to construct...” – this implies you tried and did not succeed, but since the strain was made, it should just say “we constructed...”

Line 388: add the word gene after *putA*.

Line 391: “Other” instead of “Another”

Line 549-550: this sentence needs rewording (e.g. the clearing caused by the phage indicated the lawn was sensitive).

Section 556-567: include more details on the cloning. It seems like the first paragraph is the construction of the complementation plasmid, while the second paragraph is construction of the deletion mutant using flanking homology. I’m not clear what the phrase “repair arm” means. What is SPC – if it is antibiotic for the CRISPR plasmid, say so. The construction of the point mutation allele is missing.

Response Letter

Manuscript ID: mSystems01476-25

Dear Editors and Reviewers:

Thank you very much for handling our manuscript entitled “***Investigation of therapeutic efficacy and resistance mechanisms of lytic phages targeting ST218 KL57 CR-hvKP.***” We sincerely appreciate the editor’s efforts and the reviewers’ constructive and insightful comments. These suggestions are highly valuable for improving the quality of our manuscript and have also provided meaningful guidance for our ongoing research.

We have carefully addressed all the comments and revised the manuscript accordingly. We truly appreciate the opportunity to revise and resubmit the work, and we hope that the revised version will meet the editor’s and reviewers’ expectations.

Below, we provide a detailed point-by-point response to each comment.

Reviewer 1#

Comment 1: *The authors make their case in their answer, but I now believe that Figure 3B should be tuned to be able to distinguish the many different conditions. An option could be to add a supplementary figure in which the survival curves of the two phages and the cocktail are separated in different panels. It would just be for the sake of clarity.*

Response: Thank you for your insightful review. We appreciate the reviewer’s thoughtful suggestion regarding the presentation of survival curves in Figure 3B. As shown in the current dataset, almost all treatment combinations—across both individual phages and the cocktail at three different MOIs—resulted in 100% survival. Only two conditions (JLBP1002 at MOI = 10 and the cocktail at MOI = 10) showed slight deviations in survival, in addition to the KP infection control group. Consequently, only four survival curves are distinguishable in the plot.

After careful consideration, and to avoid unnecessary duplication of curves that overlap at 100% survival, we elected to retain the current form of Figure 3B, which accurately reflects the experimental results without introducing

redundant or visually overloaded graphs. We sincerely thank the reviewer for this constructive comment.

Reviewer 2#

Comment 1: Line 284: “we attempted to construct...” – this implies you tried and did not succeed, but since the strain was made, it should just say “we constructed...”

Response: Thank you for your detailed comment. We appreciate you pointing this out. We have adjusted the expression here to read: “we successfully constructed a point-mutant strain”. Line 284.

Comment 2: Line 388: add the word gene after *putA*. Line 391: “Other” instead of “Another”

Response: Thank you for your careful comment. We have revised both of these expressions to make the expression more fluent. Line 388 and 391.

Comment 3: Line 549-550: this sentence needs rewording (e.g. the clearing caused by the phage indicated the lawn was sensitive).

Response: Thank you for your insightful review. We replaced the description of the phage sensitivity results in the spot assay, adjusting it to: “Then incubated at 37°C for 4-6 h, the clear plaque caused by phage on the lawn as sensitivity.” Line 549-550.

Comment 4: Section 556-567: include more details on the cloning. It seems like the first paragraph is the construction of the complementation plasmid, while the second paragraph is construction of the deletion mutant using flanking homology. I’m not clear what the phrase “repair arm” means. What is SPC – if it is antibiotic for the CRISPR plasmid, say so. The construction of the point mutation allele is missing.

Response: Thank you for your constructive and insightful comment. We acknowledge that the description of the strain construction methods was

confusing in the resubmission, and we have now thoroughly revised this section for clarity. The term “repair arm” refers to the homologous repair fragment used for CRISPR/Cas9-mediated genome editing. This fragment is a linear PCR product containing the fused upstream and downstream homology arms, which serves as the donor DNA for homology-directed repair and replaces the *putA* coding region following recombination. We have replaced the term “repair arm” with “repair fragment” throughout the Methods to prevent ambiguity. Line 560.

SPC refers to spectinomycin, which is the antibiotic resistance marker carried by the sgRNA plasmid used in our CRISPR/Cas9 system. Therefore, Kan/SPC plates were used to simultaneously select for cells harboring both the Cas9 plasmid (kanamycin resistance) and the sgRNA plasmid (spectinomycin resistance). Line 561.

We regret that the construction of the point-mutant strain was insufficiently described in the previous version. This has now been added to the Methods section. Briefly, the point mutation was generated using an allelic exchange strategy based on a *sacB*-containing suicide plasmid.” Line 565-575.

We would like to thank the referee again for taking the time to review our manuscript and for providing constructive and insightful comments that have substantially improved the clarity and rigor of our work. We have carefully revised the manuscript in response to each point raised. We also sincerely appreciate the editor’s efforts and assistance throughout the handling of this manuscript. We remain happy to provide any additional data, analyses, or clarifications the editors or reviewers may request, and we look forward to receiving a positive response.

With the upcoming holiday season, we would like to sincerely wish the Editor and all reviewers a Merry Christmas and a wonderful New Year.

Yours sincerely,

Liuqing Dou

December 15, 2025

Research Institute of General Surgery, Jinling Clinical Medical College,
Nanjing University of Chinese Medicine, Nanjing, China

Re: mSystems01476-25R1 (**Investigation of the therapeutic efficacy and resistance mechanisms of lytic phages targeting ST218 KL57 CR-hvKP**)

Dear Prof. Jianan Ren:

Your manuscript has been accepted, and I am forwarding it to the ASM production staff for publication. Your paper will first be checked to make sure all elements meet the technical requirements. ASM staff will contact you if anything needs to be revised before copyediting and production can begin. Otherwise, you will be notified when your proofs are ready to be viewed.

Sincerely,
Dustin Hancks
Editor
mSystems